# Erosion and weathering in carbonate regions reveal climatic and tectonic drivers of carbonate landscape evolution

Richard Ott[1,2]*, Sean F. Gallen[3], David Helman[4,5]

[1]Department of Earth Sciences, ETH Zurich, Zurich, Switzerland
[2]Earth Surface Geochemistry, GFZ German Research Centre for Geosciences, Potsdam, Germany
[3]Department of Geosciences, Colorado State University, Fort Collins, US
[4]Department of Soil and Water Sciences, Institute of Environmental Sciences, Faculty of Agriculture, Food and Environment, Hebrew University of Jerusalem, Rehovot, Israel
[5]The Advanced School for Environmental Studies, The Hebrew University of Jerusalem, Jerusalem, Israel

*Correspondence to*: Richard F. Ott (richard.ott@gfz-potsdam.de)

**Abstract.** Carbonate rocks are highly reactive and can have higher ratios of chemical weathering to total denudation relative to most other rock types. Their chemical reactivity affects the first-order morphology of carbonate-dominated landscapes and their climate sensitivity to weathering. However, there have been few efforts to quantify the partitioning of denudation into mechanical erosion and chemical weathering in carbonate landscapes such that their sensitivity to changing climatic and tectonic conditions remains elusive. Here, we compile bedrock and catchment-average cosmogenic calcite-$^{36}$Cl denudation rates and compare them to weathering rates derived from stream water chemistry from the same regions. Local bedrock denudation and weathering rates are comparable, ~20 – 40 mm/ka, whereas catchment-average denudation rates are ~2.7 times higher. The discrepancy between bedrock and catchment-average denudation is five times lower compared to silicate-rich rocks illustrating that elevated weathering rates make denudation more spatially uniform in carbonate-dominated landscapes. Catchment-average denudation rates correlate well with topographic relief and hillslope gradient, and moderate correlations with runoff can be explained by concurrent increases in weathering rate. Comparing denudation rates with weathering rates shows that mechanical erosion processes contribute ~50% of denudation in southern France and ~70% in Greece and Israel. Our results indicate that the partitioning between largely slope-independent chemical weathering and slope-dependent mechanical erosion varies based on climate and tectonics and impacts the landscape morphology. This leads us to propose a conceptual model whereby in humid, slowly uplifting regions, carbonates are associated with low-lying, flat topography because slope-independent chemical weathering dominates denudation. In contrast, in arid climates with rapid rock uplift rates, carbonate rocks form steep mountains that facilitate rapid, slope-dependent mechanical erosion required to compensate for inefficient chemical weathering and runoff loss to groundwater systems. This result suggests that carbonates represent an end-member for interactions between climate, tectonics, and lithology.

## 1    1 Introduction

Landscapes evolve through a combination of mechanical erosion processes and the chemical breakdown of minerals, hereafter referred to as erosion and weathering, respectively (e.g., Gabet and Mudd, 2009). In most silicate-rich landscapes, denudation

– the sum of physical erosion and chemical weathering – is dominated by erosion processes (Larsen et al., 2014). In contrast, carbonates are more susceptible to weathering (Gaillardet et al., 1999), likely resulting in a different partitioning of denudation into erosion and weathering for the same climatic and tectonic conditions (Ott et al., 2019). These differences in denudation partitioning should have pronounced effects on landscape morphology, presumably making carbonate landscapes more sensitive to differences in climate than silica bedrock and altering surface and groundwater pathways (e.g., karst hydrology). However, few studies have quantified the relative contributions of erosion and weathering in carbonate-dominated landscapes, the effect of climate and tectonics on such partitioning, and its impact on topography.

The limited number of studies investigating denudation partitioning in carbonates stems largely from challenges in quantifying long-term erosion. Weathering can be constrained by measuring solute fluxes from dissolved loads and direct outcrop measurements with limestone tablets (e.g., Calmels et al., 2014; Goodchild, 1890; Plan, 2005). Denudation, the sum of erosion and weathering, can be quantified using cosmogenic radionuclides (CRNs). These CRNs are produced close to the Earth's surface, and their concentration in surface minerals is inversely proportional to the denudation rate (Lal, 1991). This technique has been widely applied to quartz-bearing rocks. Recent advances in cosmogenic $^{36}Cl$ production rate calibration and calculation (Marrero et al., 2016b; Schimmelpfennig et al., 2009) now allow for accurate calculation of denudation rates in carbonates at the outcrop and catchment scale. Through the comparison of CRN measurements with weathering rates, it is possible to isolate the contribution of mechanical erosion by subtracting dissolved load-derived weathering from denudation rates (Blanckenburg et al., 2004; Dixon and Blanckenburg, 2012; Ott et al., 2019), provided that the disparate integration timescales (days to years for weathering rates, thousands of years for CRNs) of each measurement have a negligible impact on measurement comparisons, and when CRN weathering biases can be accounted for (Ott et al., 2022; Riebe et al., 2001a). Studies applying a combination of CRN denudation rates and elemental analysis, e.g., measuring concentrations of immobile elements in the bedrock and regolith/saprolite have determined denudation portioning in silicic landscapes (Ferrier et al., 2012; Riebe et al., 2001b, 2003, 2004). However, these combined approaches have not been extended to carbonate landscapes, yet.

Studies using CRN or solute flux measurements have arrived at different conclusions regarding carbonate denudation. Several recent studies assumed that carbonate erosion is negligible even in mountainous landscapes (Avni et al., 2018; Ryb et al., 2014b, 2014a), implying that weathering dominates carbonate denudation budgets. However, other studies found that erosion processes likely play a critical role in carbonate denudation (Covington et al., 2015; Newson, 1971; Ott et al., 2019; Thomas et al., 2017a). Here, we contribute to this growing effort by compiling and comparing available cosmogenic calcite-$^{36}Cl$ denudation rate measurements with catchment-average weathering rates collected from the same areas to quantify the partitioning between erosion and weathering in carbonate terrains across climatic and tectonic gradients. We use this analysis to illustrate how denudation partitioning varies as a function of climate and tectonics and highlight that carbonate regions are more susceptible to climate-topography interactions than silicate-rich rocks, thus offering the unique potential to identify climate signatures in landscape evolution.

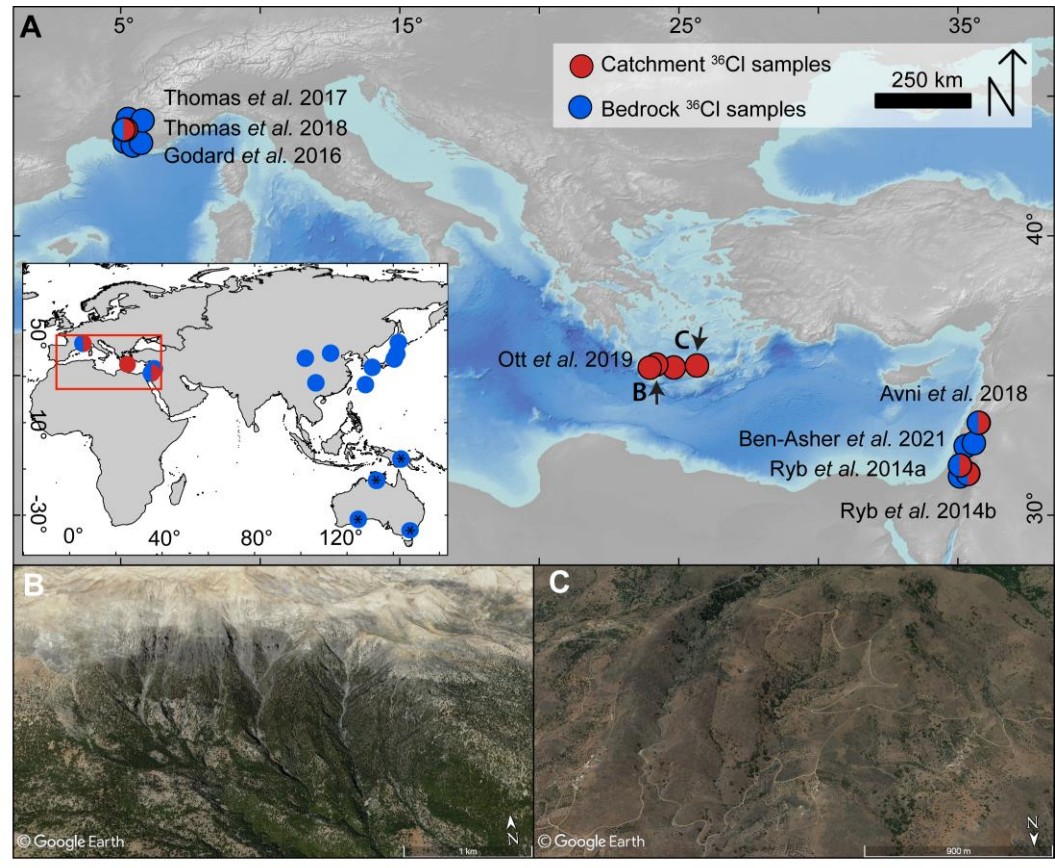

**Figure 1: (A)** Location of [36]Cl denudation rate measurements compiled for this study in the Mediterranean and globally (inset). *samples not included due to lack of location and compositional data. **(B)** Typical steep carbonate catchment in the Lefka Ori range on Crete, Greece, with partial forest cover and small landslide scars. **(C)** Typical low relief catchment on Crete.

## 2 Methods

### 2.1 [36]Cl denudation rate compilation

We compiled 232 bedrock and 43 alluvial sediment [36]Cl measurements from the literature to determine denudation rates at the outcrop and catchment scale, respectively (Fig.1, Tab. S1). Measurements from alluvial samples were only available from studies around the Mediterranean (Fig. 1). The bedrock denudation rates were sampled from bedrock outcrops or amalgamated clasts on hillslopes or ridgetops, thus recording local bedrock denudation rates. In contrast, the alluvial sediment samples are assumed to be well-mixed and provide a catchment-average denudation rate. Published bedrock denudation rates exist for Eastern Asia and the Mediterranean, spanning a range of climate zones from the arid Negev desert in Israel to the humid mountains of Japan, with mean annual precipitation (MAP) ranging between 190 and 2300 mm/a. Alluvial catchments span a narrower range of precipitation rates between 200 and 1100 mm/a and drainage areas ranging between a few km² in southern France to tens of km² in Israel and Crete, Greece (see Fig. S1 for catchment maps). Most of the samples in the compilation are

from relatively pure, un- to moderately metamorphosed, massive, and bedded limestone bedrock with varying amounts of dolomite (Avni et al., 2018; Godard et al., 2016; Ott et al., 2019; Ryb et al., 2014a, 2014b; Thomas et al., 2017b, 2018; Xu et al., 2013). However, the bedrock samples also contain marbles (Matsushi et al., 2010), pure dolostones and chalk (Ben-Asher et al., 2021), and samples of unknown carbonate lithology (Yang et al., 2020).

The calculation of denudation rates from $^{36}$Cl concentrations requires knowledge of the chemical composition of the host rock, and the so-called target mineral analysed (Schimmelpfennig et al., 2009). To compare denudation rates from different studies, we compiled all chemical sample data from the literature or, if unavailable, contacted the authors (Tab. S1). Subsequently, all denudation rates were calculated with CRONUScalc v2.1 (Marrero et al., 2016a). Chemical weathering can bias cosmogenic nuclide-derived denudation rates. Weathering in regolith can overestimate $^{36}$Cl denudation rates because the soluble target mineral calcite will have a shorter regolith residence time than the bulk regolith due to its high solubility. We use the methods developed by Ott et al. (2022) to correct all alluvial denudation rates for regolith weathering. However, the difference between weathering-corrected denudation rates and standard CRONUScalc calculation is generally < 5% (Tab. S2). Due to the uncertainties in bedrock mineralogy required for accurate weathering corrections, and their low impact on the denudation rate calculation, we present the uncorrected denudation rates in the main text. For more information on the weathering correction of denudation rates, we refer the reader to the supplemental text.

## 2.2 Carbonate weathering rate calculations

We calculated carbonate weathering rates for areas with published catchment-average denudation rates to infer the landscape-scale erosion. Erosion can be assumed to equal denudation minus weathering because, despite deep solution features such as caves, volumetrically, most carbonate dissolution occurs close to Earth's surface (Ford and Williams, 2010; Gunn, 2013). Field studies measuring water chemistry at different depths below the surface show that the most of weathering occurs within the first meters below the Earth's surface (Gunn, 1981; Williams and Dowling, 1979). Furthermore, studies quantifying the volumetric percentage of voids in carbonate bedrock found that only 0.003-0.5 % of the karst volume has been removed by deep dissolution (Worthington and Smart, 2004). Hence, we assume weathering rates primarily reflect near-surface mass removal. In southern France, we used time-averaged water data of $[Ca^{2+}]$ and $[Mg^{2+}]$ for springs and wells, provided by the national portal of water data (ADES) (Tab. S3) (Ott et al., 2019). On average, five measurements were available per site, showing only minor variations in $[Ca^{2+}]$ and $[Mg^{2+}]$ concentration (Tab. S3). Therefore, we assume chemostatic behaviour of the carbonate water sources as has been observed for carbonate regions on global and local scales (Bluth and Kump, 1994; Gaillardet et al., 2018; Ott et al., 2019; Zhong et al., 2017). For the catchments in Israel, regional well and spring water chemistry data were published by Ryb *et al.* (2014a), and in Crete, weathering rates were reported by Ott *et al.* (2019). All chemical data were corrected for precipitation input by assuming that all [Cl-] is derived from precipitation, and that other cations are scaled by seawater ratios (Stallard and Edmond, 1981). Water data from Israel were not corrected due to unknown [Cl-]; however, the correction for $[Ca^{2+}]$ was usually < 1% and is therefore assumed to be negligible.

To estimate the water flux, we used satellite-derived precipitation and actual evapotranspiration data (AET). Precipitation was derived from the WorldClim 1 km dataset (Fick and Hijmans, 2017) and averaged for each catchment. To determine the runoff available for dissolution, AET was estimated from a parameterization of vegetation indices (PaVI-E)

model (Helman et al., 2015) using MODerate resolution Imaging Spectroradiometer (MODIS) satellite data from 2000 to 2016 at 1 km resolution and water vapor flux data from the eddy covariance tower international net (FLUXNET). PaVI-E was successfully validated against basin-scale AET derived from water balance calculations (i.e., AET= P – Q) in the Eastern Mediterranean region ($R^2 = 0.85$, $p<0.05$) and was shown to be comparable with other well-established physically-based AET models (Helman et al., 2015, 2017a). It has been used to study climate change impacts on the terrestrial water cycle and is

considered a reliable tool for water balance calculations (Helman et al., 2017c, 2017b).

The weathering rate W in mm/ka was then calculated by assuming that all $[Ca^{2+}]$ and $[Mg^{2+}]$ (mg/l) in the water are derived from carbonate dissolution and using the water flux from our local runoff calculation (mm/ka) for each catchment, with the following equation

$$(1) \quad W = \frac{\left([Ca] + \frac{[Ca]}{M_{Ca}} * M_{CO_3}\right) * (MAP - AET)}{\rho_{Calcite}} + \frac{\left([Mg] + \frac{[Mg]}{M_{Mg}} * M_{CO_3}\right) * (MAP - AET)}{\rho_{Dolomite}}$$

with $\rho_{Calcite}$ and $\rho_{Dolomite}$ of 2.5 and 2.85 g/cm³, respectively. We chose to express the weathering rate as a surface lowering rate with units of [L T$^{-1}$], because this is the more common way to quantify denudation rates from cosmogenic nuclides. For groundwater water samples, we estimated a recharge area, equivalent to a groundwater catchment, for runoff averaging based on surface topography and local geology (see Fig. S2 for estimated groundwater recharge areas). Weathering rates were calculated with an uncertainty of 10% on the precipitation data and 20% on the actual evapotranspiration. Because

recharge areas of springs and wells in karstic terrains can deviate significantly from topographically estimated areas, we calculated a second estimate based on the P and AET average of the entire mountain range where the spring or well is located (Tab. S3). However, the difference in the average dissolution rate between both approaches is < 3%, and we, therefore, choose to use the topographic recharge area estimates outlined in Fig. S2.

## 2.3 Topographic and climatic metric calculation

Topographic metrics were measured on a 1 arc-second (~30 m) Shuttle Radar Topography Mission dataset. Slope, mean elevation, mean slope, and mean local relief (500 m radius) were calculated using TopoToolbox (Schwanghart and Scherler, 2014). Analogous to the carbonate weathering rate calculation, mean annual precipitation (MAP) for catchment-average $^{36}$Cl-samples was calculated by averaging WorldClim data (Fick and Hijmans, 2017). For locations with catchment-average denudation rates, we used our AET estimates to calculate the specific runoff in mm/a by subtracting MAP from AET values.

# 3    Results

## 3.1    Rates of carbonate denudation

Local bedrock denudation rates, as well as weathering rates calculated from water data, are similar and generally fall between 20 - 40 mm/ka (mean ± 1σ: 29 ± 25 mm/ka and 32 ± 19 mm/ka, respectively) (Fig. 2). In contrast to the local bedrock denudation rates, catchment-average denudation rates from alluvial samples are ~2.7 times higher than weathering rates (mean 81 ± 35 mm/ka) (Fig. 2). A Mann-Whitney U test shows that distributions of weathering, bedrock, and catchment-average rate are statistically different at > 99 percent confidence (Fig. 2A). The trends from the global data set, can also be observed on the local scale, where climatic and lithologic conditions are more uniform (Fig. 2B, C). In the Soreq catchment in Israel, and the carbonate mountain ranges of southern France, bedrock and weathering rates are similar, and catchment-average denudation rates are substantially higher.

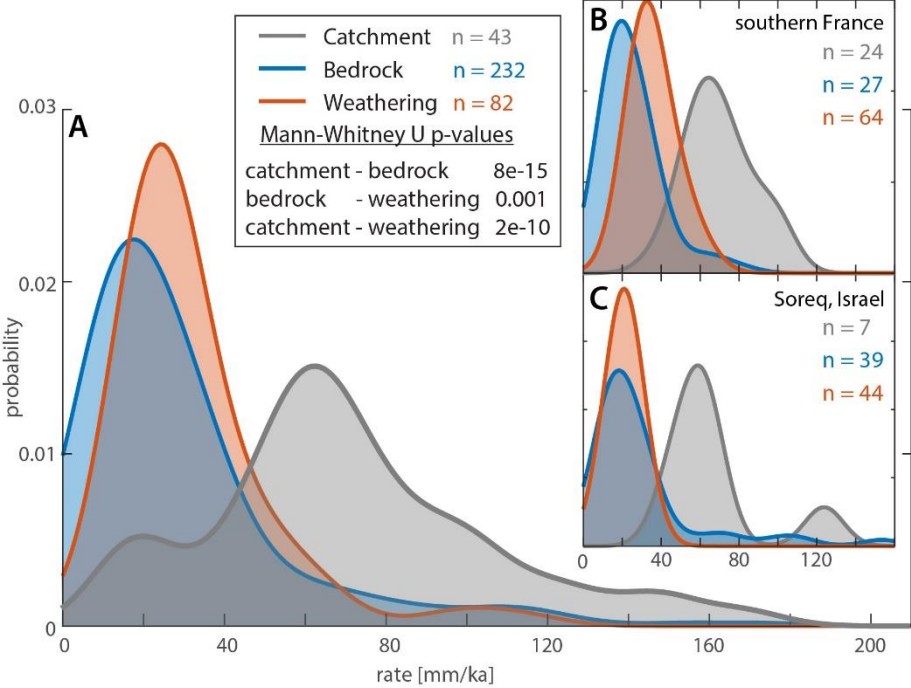

**Figure 2: (A) Kernel density plot of weathering, bedrock, and catchment-average denudation rates in carbonate catchments for all compiled data. The legend also displays the results from the nonparametric Mann-Whitney U tests, where a low p-value indicates a difference in the median value. (B) & (C) The same plots for southern France and the Soreq catchment in Israel; the only two regions where all three data types are available.**

## 3.2    Carbonate denudation and weathering rates as a function of topographic and climatic factors

Correlations between local bedrock denudation rate and topographic and climatic variables are weak (Pearson's r < 0.25) (Fig. 3). Some region-specific local bedrock denudation rates show positive relationships with mean annual precipitation (MAP) (Fig. 3D). However, the covariation of topographic metrics and precipitation (Pearson's r generally > 0.5, Tab. S5,6) associated

with orography impedes isolating climatic effects from these correlations. Unfortunately, AET data were not available for
several bedrock sampling locations, and therefore no comparison with runoff could be performed.

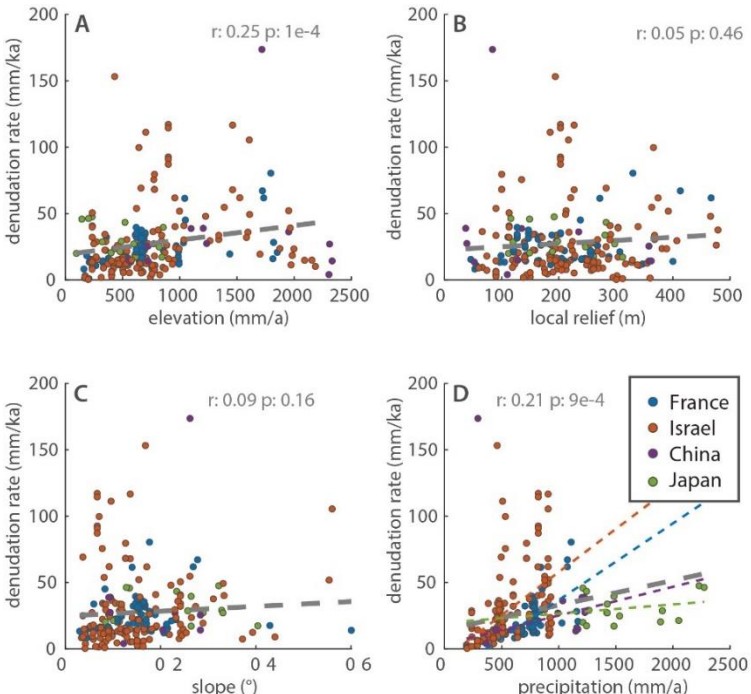

**Figure 3: Correlation between $^{36}$Cl bedrock denudation rates, topographic metrics and mean annual precipitation based on linear regression. Colours indicate different the sample location; Pearson's r values and p-values for the regressions are displayed. In panel (D), region-specific positive relationships with mean annual precipitation (MAP) exist and are highlighted.**

Catchment-average denudation rates show strong-to-moderate correlations with topographic metrics such as local relief and slope (Pearson's r of 0.65 and 0.5, respectively, Fig. 4A, B). Catchment-average rates also scale strongly with precipitation; however, all study sites exhibit orographic precipitation gradients such that precipitation is correlated with topography (Fig. 4C inset). However, when comparing denudation rates to runoff, the correlation is weak (r = 0.38). The slopes of the regression lines for catchment-average denudation and weathering rates are similar (Fig. 4D). Weathering rates correlate weakly-to-moderately with topographic and climatic variables but remain consistently lower than catchment-average denudation rates across all variable gradients. The same trends can also be observed on a local scale. Crete is the only region where samples cover significant gradients of all analysed metrics, and generally the correlations are similar to Fig. 4 (Fig. S3).

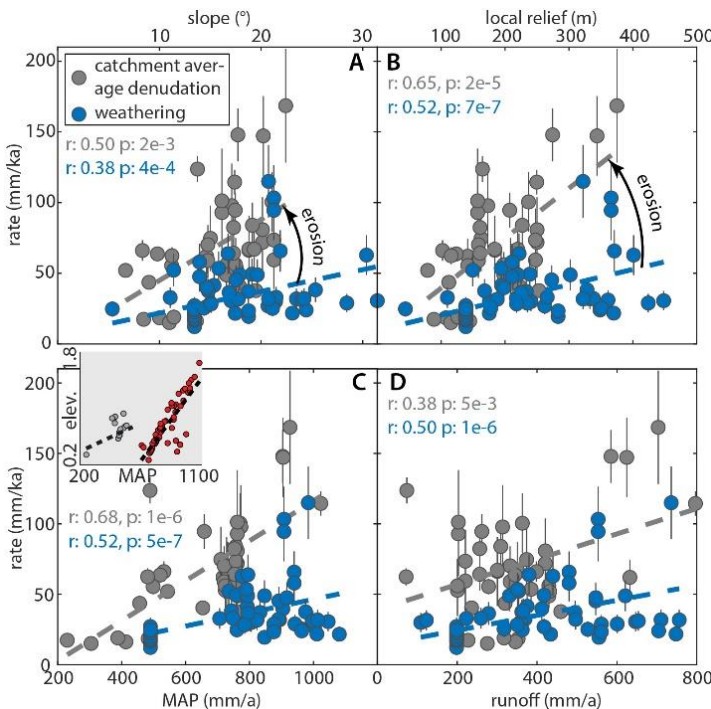

**Figure 4:** Correlations between catchment average [36]Cl denudation rates, carbonate weathering rates, and topographic and climatic metrics based on linear regression. Values of correlation coefficient and p-value displayed for catchment-average denudation and weathering rates. Inset-C: Correlation between mean annual precipitation (MAP) and elevation illustrating orographic precipitation in the analysed areas (Israel, grey; France and Crete, red).

## 4    Discussion

### 4.1    Rates of carbonate bedrock denudation

The similarity of denudation rates for outcrops and weathering suggests that weathering may be the dominant denudation process for locally exposed bedrock on hillslopes. This has also been suggested based on the observations of outcrops with micro erosion meters (Cucchi et al., 1996; Furlani et al., 2009). But due to the difference in the spatial scale of catchment weathering rates and local bedrock denudation rates, this link remains vague. The large scatter in the relationship between bedrock denudation rates and topographic and climatic variables is likely related to a number of site-specific factors that cannot be accounted for in a large-scale compilation. Though all samples are from carbonate bedrock, likely significant variations in mineralogy, diagenesis, and bedding structure exist, which may have a strong influence on the resulting denudation rate. Additionally, the topographic variables were measured at 30 m resolution, which only gives a rough estimate of local slope for a bedrock sample. The exact geomorphic position of bedrock samples is likely to be very important for the measured denudation rate and cannot be resolved within this data set. Moreover, the at-a-site variations in bedrock denudation rates are commonly large (e.g., Avni et al., 2018; Ryb et al., 2014a), suggesting that small-scale variability is high and hinders the

observation of clear trends. Thus, our further analysis of climatic and tectonic controls focuses primarily on the catchment-average samples.

## 4.2   Discrepancy between bedrock and catchment-wide denudation rates

Analogous to our findings, a discrepancy between bedrock and catchment-average denudation rates has also been documented in a compilation of silicate-rich rock units (Portenga and Bierman, 2011). This difference has been interpreted as reflecting growing topographic relief because rivers are thought to erode faster than the ridgelines (Hancock and Kirwan, 2007; Small and Anderson, 1998; Thomas et al., 2017a), or higher weathering rates below soil-covered bedrock and therefore a sampling bias toward fresh outcrop surfaces (Portenga and Bierman, 2011). It seems unlikely that all studies comparing bedrock-interfluve samples and catchment averages would find increasing relief and therefore another explanation is needed. If the rate discrepancy is due to soil cover, measured soil production rates should equal the catchment-average rates. Yet, a global compilation of soil production rates shows that most measurements are significantly lower than the catchment-average denudation rates (Heimsath et al., 2012). An alternative explanation is that both local bedrock denudation and soil production rates, whether in carbonates or siliciclastic rocks, represent a biased sampling of stable portions of the landscape (e.g., areas not affected by recent mass-wasting). Catchment-average sampling incorporates all portions of the upstream landscape and reflects a mix between the locally high denudation rate associated with mass wasting and a "background" rate set by bedrock denudation and soil production.

In silicate-rich regions, catchment-average denudation rates are between 10 (median) and 20 times (mean) higher than bedrock denudation (Portenga and Bierman, 2011). The smaller ~2.7-fold discrepancy between local bedrock and catchment-average rates in carbonate terrains is best explained by the elevated role of chemical weathering in carbonates (>1/3 of denudation, Fig. 3) compared to silica-rich rocks (< 5% of denudation) (Larsen et al., 2014). This result implies that carbonate regions have more uniform lowering rates due to weathering compared to silicate-rich landscapes. Below we discuss how these findings have important implications for landscape evolution in carbonates and their topographic response to tectonic uplift and climate because erosion is slope-dependent, whereas weathering mostly depends on climate and vegetation (Gaillardet et al., 2018).

## 4.3   Mechanical erosion in carbonate landscapes

We consider three possible explanations for the two-to three-fold difference between catchment-scale weathering rates and catchment-average denudation rates: (1) Catchment-average rates overestimate denudation due to low [36]Cl concentration material from deep landslides (>5 m, Yanites et al. 2009); (2) higher runoff earlier in the averaging time window of cosmogenic nuclides (Ryb et al., 2014a, 2015); (3) mechanical denudation of carbonates. A deep-seated landslide bias seems unlikely due to the consistent discrepancy between catchment-average denudation and weathering in a diverse array of topography, from low relief and slope areas in Israel to the rugged mountainous terrain of Crete (Figs. 1, 2). Furthermore, most sampled

catchments drain moderately sized areas (10's of km$^2$) and are therefore large enough to buffer potential biases arising from mass wasting (Niemi et al., 2005; Schide et al., 2022; Yanites et al., 2009).

The disparate integration timescales of cosmogenic nuclide (10$^2$– 10$^5$ yrs) and weathering rate measurements (0 – 10$^2$ yrs) lead Ryb et al. (2014a; 2015) to suggest that higher CRN-derived catchment-average denudation rates integrated a history of more vigorous weathering associated with past wetter climates. Most of the compiled catchment-average denudation rates reflect time-integrated rates from the modern to mid-Holocene, with the lowest rates integrating timescales to the last glacial maximum (LGM). To test this hypothesis, we compared current precipitation to 1 km resolution WorldClim paleo-precipitation maps from three different climate models (Fick and Hijmans, 2017). We found that precipitation was 2 to 27% higher in the mid-Holocene compared to the modern era and between 25% drier or 55% wetter in the LGM, depending on the area and preferred model in the three Mediterranean sampling areas (Fig. 5). Since weathering scales linearly with the water availability (White, 1984), these changes in precipitation, even with a total drawdown of actual evapotranspiration, cannot explain the data discrepancy through an increased paleo-water flux when integrated over time.

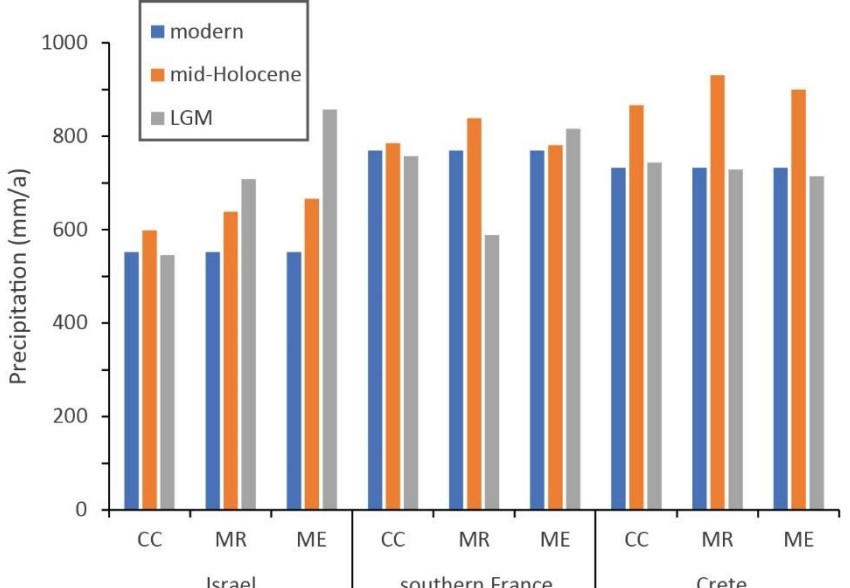

**Fig.5: Comparison of modern and paleo-precipitation rates from climate models for all areas with published catchment average denudation rates. For three different WorldClim paleo-climate models (CC, MR, ME)** (Fick and Hijmans, 2017)**, paleo-precipitation values were extracted for all sampled** [36]**Cl sampling locations, averaged for each region and compared to modern precipitation rates. LGM – Last Glacial Maximum**

Overall, these observations suggest that in Mediterranean carbonate landscapes, mechanical erosion contributes significantly to denudation because (1) catchment-average denudation rates are substantially higher than weathering rates, (2) denudation rates scale with topographic steepness, which is expected for gravity-driven erosion processes, (3) the difference between denudation and weathering, which we interpret as erosion, also increases with increasing topographic steepness (Fig.

4), whereas (4) the parallel trends of denudation and weathering with runoff suggest that the scaling of denudation with runoff can be accounted for by increased weathering (Fig. 4D).

## 4.4    Landscape evolution in carbonates compared to other rock types

Averaging catchment-average denudation and weathering rates for southern France, Israel (Soreq), and Crete, we find erosion-to-weathering ratios of 1.0, 2.2, and 2.5, respectively (Fig. 6). The substantial amount of erosion in these carbonate landscapes requires steep, high relief topography allowing slope-dependent processes to thrive. Carbonate weathering rates are largely dependent on water flux, temperature, and vegetation (Atkinson and Smith, 1976; Gaillardet et al., 2018) and can therefore be seen as a proxy for climate. Denudation rates typically evolve to balance tectonic uplift and can be viewed as a tectonic proxy. Carbonates in slowly uplifting areas will denude sluggishly. Under favourable climatic conditions, weathering alone is sufficient to balance rock uplift with little need for steep topography to develop to enhance erosion rates. This conceptual model explains why carbonates remain topographically subdued under certain conditions. Less-soluble lithologies will stick out because surface-lowering can only be achieved through erosion requiring local slopes to form. This end-member can be observed in Ireland or the Appalachians, where carbonates denude mostly through dissolution, and eroding sandstone ridges form the topographic highs (Duxbury et al., 2015; Simms, 2004), though some of the topographic features in Ireland may have been overprinted by Quaternary glaciations.

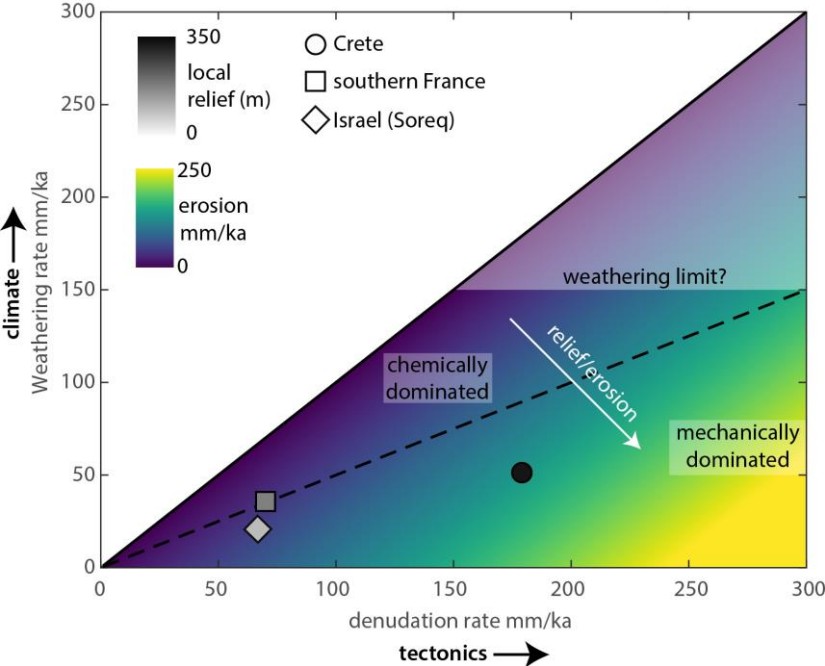

**Figure 6: Mean weathering versus mean denudation rates for the sites with alluvial denudation data. The dashed line is the 1:1 ratio of weathering and erosion. Weathering-dominated landscapes will tend to be subdued; high erosion rates will lead to high relief areas dominated by slope-dependent mechanical processes.**

The local climate will mainly set the weathering rate. Still, it may be subject to a dissolution speed limit because of (1) a chemical threshold of <200 mg/l water hardness, which has been observed globally (Covington et al., 2015; Gaillardet et al., 2018), (2) an increase in precipitation will typically be partially compensated by an increased AET, and (3) tropical regions with high runoff have lower water hardness due to a decrease in carbonate solubility with temperature (Atkinson and Smith, 1976; Gaillardet et al., 2018). A global compilation of carbonate weathering rates finds a maximum rate of ~140 mm/ka (Gaillardet et al., 2018), similar to our highest rate of 115 mm/ka. These rates may represent a carbonate weathering speed limit beyond which local uplift cannot be accommodated; this is when erosion becomes dominant and topography steepens (Fig. 6). Erosion is thought to accelerate weathering by breaking rocks down and increasing surface area (e.g., Rinder and von Hagke, 2021), resulting in a direct feedback between erosion and weathering. In silicate-rich rocks, this has been shown to be true for low and moderate denudation rates, with a transition to a climate limit at high denudation rates (Bufe et al., 2021; West et al., 2005). However, some observations suggest that the feedback between erosion and weathering rate is weak or nonexistent in carbonates. Long-term weathering rates of carbonates in experiments were independent of grain size and texture (Levenson et al., 2015). Additionally, water chemistry measurements that suggest carbonate weathering is generally limited by water and acid availability (Bufe et al., 2021), except for regions where carbonates are a minor component of the bedrock or denudation increases the acid supply (Bufe et al., 2021; Erlanger et al., 2021).

In carbonate regions where mechanical erosion prevails, the formation of karst features will cause surface water infiltration and lower the discharge of surface streams. Reduction in stream discharge will decrease the erosional efficiency and cause steepening of the landscape compared to regions with less-soluble bedrock (Ott et al., 2019). This explains why carbonate terrain in locations with arid to semi-arid climates and significant uplift rates form steeper topography than areas underlain by silicate-rich lithologies (Ott, 2020; Ott et al., 2019), and carbonates form the low parts of the landscape, e.g. in Ireland or the Appalachians, where uplift rates are low, and the climate is favourable for weathering (Gallen, 2018). Undoubtedly, feedbacks may exist where mountain building causes more orographic precipitation, as seen in the Mediterranean study sites, which increases weathering. Therefore, the climate and tectonic axis in Figure 6 should not necessarily be seen as entirely independent. Additionally, Erlanger et al. (2021) showed that significant amounts of dissolved load could reprecipitate as secondary calcite in rivers flowing through carbonates, as soil pore water equilibrated to high $CO_2$ levels degasses upon entering streams. This effect has not been observed in the ephemeral catchments studied here but shows how in wetter regions erosion and weathering can be linked, and catchment and local scale weathering rates may differ.

Our results have important implications for landscape evolution. The difference in denudation partitioning compared to silicate-rich rocks makes the topography of carbonate regions more sensitive to the interplay of tectonics and climate. Studies investigating the climatic and tectonic effects on topography typically report that tectonics govern the first-order landscape morphology (Seybold et al., 2021). However, in mountain belts with a lot of exposed carbonate bedrock, the topographic expression could be different from silicate-dominated mountains. Indeed, studies of the relationship between bedrock channel steepness and erosion rate or rock uplift rate show that the functional form of these parameters is different between silica- and

carbonate-rich bedrock. Studies of bedrock rivers in silica-rich rock generally show channel steepness – erosion rate relationships are described by a power function with an exponent *n* between ~1.5 – 3 (DiBiase et al., 2010; Harel et al., 2016; Ouimet et al., 2009). In contrast, similar studies in carbonate-rock bedrock channels find this exponent *n* is ~0.5 (Attal et al., 2011; Gallen and Wegmann, 2017; Royden and Taylor Perron, 2013; Whittaker et al., 2008). The implication of these results is that as channels steepen, bulk erosional efficiency increases in silicia-rich bedrock, whereas in carbonate-rich bedrock, it declines. These contrasting behaviours are perhaps related to surface water infiltration associated with the high reactivity of carbonates relative to silica-rich rock, but the exact causal mechanisms are currently unknown (Gallen and Wegmann, 2017).

These behaviours and the sensitivity to environmental variables could be investigated by studying topography, together with erosion and weathering rates in carbonate regions along climatic and tectonic gradients. Such studies could benefit from the recently developed [10]Be-[36]Cl paired nuclide approach by Ott et al. (2022) that can be used to determine long-term erosion and weathering rates from the same sample. We also speculate that the chemical threshold of carbonate weathering and runoff loss due to surface water infiltration in carbonate regions, might cause non-linear topographic responses to climate and tectonic forcing. Many of these hypotheses can be tested with future studies that fill the climate-tectonic parameter space in figure 6. Hence, we argue that the reactive nature of carbonate regions offers the potential to observe strong controls of climate responses to tectonics on landscape evolution, and carbonate-dominated mountain belts should be targeted in future work.

## 5    Summary and Conclusions

We compiled all available [36]Cl denudation rate measurements from carbonate regions and compared them to weathering rates. Our main findings are:

(1) Bedrock denudation and weathering rates range are similar and range between 20-40 mm/ka, suggesting that weathering may be the main process of lowering at the bedrock sampling scale. Catchment-wide denudation rates are on average 2.7 times higher, resembling the incorporation of all denudation processes in the catchment.

(2) The discrepancy between bedrock and catchment-wide denudation rates in carbonates is 5 times lower than in silicate-rich rocks, reflecting a higher weatherability and suggesting more uniform landscape lowering.

(3) Denudation rates are higher than weathering rates, and the differences increase with topographic steepness. This suggests substantial mechanical erosion in carbonate regions, which we estimate to be between 50-70% of denudation in the sampled Mediterranean regions.

(4) In regions where uplift rates are low, denudation is mostly achieved through weathering, carbonates will form the low part of the landscape, because no slopes need to be formed for denudation. However, in regions of higher uplift, erosion needs to occur, which requires slopes to form. In these regions, infiltration commonly lowers the efficiency of surface processes, thereby steepening the carbonate parts of the landscape. Hence, depending on the partitioning

of denudation into weathering and erosion, carbonates form either the steep or the low part of landscape compared to silicate-rich rocks.

## 6    Data availability

All data required to reproduce the study are reported within the main text and supplemental files.

## 7    Author Contributions

R.F.O.: Conceptualization, Funding acquisition, Investigation, Data Curation, Writing – original draft preparation. S.F.G: Conceptualization, Funding acquisition, Writing - review & editing, D.H.: Methodology, Writing - review & editing.

## 8    Competing Interests

The authors declare no competing interests.

## 9    Acknowledgments

R.O. was funded by Marie Curie actions grant number 674899 and SNSF grant number P2EZP2_191866. S.F.G. was partially supported by NSF awards EAR-1945970, EAR-2041910, and EAR-2139894 and funds provided by Colorado State University.

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
