# Peer review of "Erosion and weathering in carbonate regions reveal climatic and tectonic drivers of carbonate landscape evolution"

_EGUsphere, 2022_

## Referee Comment (RC2)

Review of *Erosion and weathering in carbonate regions reveals climatic and tectonic drivers of carbonate landscape evolution* submitted to EGUsphere

I found this paper to be a clearly written and illustrated compilation of chemical weathering and denudation rates in carbonate landscapes. The data compilation and analysis is timely, and the interesting discussion is clearly relevant to understanding landscape evolution under different tectonic and climate boundary conditions. Overall, the manuscript is in a great shape, and I only have a few comments that I suggest to consider before publication.

**Calculation of weathering rate**

It would help me if you could state explicitly (i.e. with equations) how you convert concentrations into "weathering rates" in mm/yr and how you interpret these rates. I presume that you need a density to get a quantity in mm/yr? Did you use densities of just Mg and Ca, or did you consider some form of calcite/dolomite?

On a minor note: To me, a denudation rate (in $L\ T^{-1}$) is an average/effective rate of surface lowering. Does the weathering rate here (in $L\ T^{-1}$) have an equivalent interpretation as a surface lowering rate? Typically, dissolution (or weathering) rates on the mineral scale are defined as a change in concentration per time ($Mol\ L^{-3}\ T^{-1}$). On the landscape scale, the term "weathering rate" is inconsistently used in the literature. In general, multiplying concentrations by runoff yields a weathering flux (in the sense of a flow of mass per unit area per unit time). Comparing denudation and weathering fluxes (in $M\ L^{-2}\ T^{-1}$) would be more intuitive for me – but either one works, as long as the calculations are clear.

**Comparison of weathering and denudation rates**

Not all study areas yield measurements of both bedrock and catchment-averaged denudation rates. I don't think it should be a big issue, but in case there were some biases in climate, lithology or tectonic regimes toward the Mediterranean samples with respect to the other samples, it could affect the distributions of the relative rates. Could you state, how much the relative distributions in Fig 2 would change if only those datasets were considered that have all data constrained?

**Discussion of weathering limits**

As far as I understand, the authors suggest that silicate weathering is sensitive to physical erosion (L250) in contrast to carbonate weathering that is limited by climate (L209). This seems a bit simplified and perhaps misleading. Increasingly, models and data suggest that silicate weathering fluxes – at least on the scale of mountain catchments – are often not sensitive to denudation and become limited by climate at relatively moderate denudation rates of $10^{-2} – 10^{-1}$ mm/y (Bufe et al., 2022; Bufe et al., 2021; Gabet and Mudd, 2009; Hilley et al., 2010; West, 2012; West et al., 2005). In turn, where carbonates are a minor component of the

rocks and waters remain undersaturated with respect to carbonates, they can be sensitive to denudation, even at very high denudation rates (Bufe et al., 2022; Erlanger et al., 2021). Even where waters are saturated with respect to carbonate, they can be indirectly limited by denudation if denudation increases the acid supply (Bufe et al., 2021; Hilton and West, 2020). I do not think that a detailed discussion of weathering limits is necessarily needed for the paper, but I would like to caution against simplifying the discussion to silicate weathering = denudation sensitive and carbonate weathering = climate sensitive.

**Bedrock versus catchment scale**

In conclusion 1, the authors suggest that weathering is dominant on the bedrock scale – because bedrock denudation rates mirror weathering rates. Perhaps I missed some information in the manuscript, but I do not follow. As far as I understand, there is data on (1) denudation on bedrock and catchment scales and (2) weathering on the catchment scale. The contrast between denudation rates on bedrock and catchment scales was assigned to different erosion processes on these spatial scales (e.g. mass movements). Is it therefore possible to compare weathering on the catchment scale to denudation on the bedrock scale? Perhaps, weathering on the bedrock scale is also slower than on the catchment scale – or maybe it is even higher. I am not sure if these rates can be directly compared – or perhaps I missed some discussion of this in the manuscript?

**Line comments**

SI: Not all data table entries have units – and not all missing units are obvious. Would be great if fixed.

L13: "Their" in this sentence refers to "carbonate-dominated landscapes" as far as I understand. What is meant by "their climate sensitivity". Vegetation, denudation, weathering, or other parameters could all be climate sensitive in the landscape.

L17: Which discrepancy is "This" referring to? Discrepancy between bedrock denudation versus catchment denudation or discrepancy between catchment denudation and catchment weathering?

L19: I do not follow how a lower discrepancy between weathering and denudation makes denudation more spatially uniform? In other words – how does a difference between two rates bear implications for the spatial distribution of these rates?

L32: To me, the word "which" seems to refer to erosion – but it should be denudation?

L65: At the beginning of the methods, it would be great to have an introductory sentence or two on the selected study areas. What range of denudation rates and climate do they span? Were all available $^{36}Cl$ areas included or were some measurements/study areas discarded? Perhaps most importantly, it would be great to have a sense of the range of lithologies. Are these all pure carbonate landscapes, or are there mixed-siliceous rocks. Are the carbonates all

unmetamorphosed massive limestones, or are there marbles or more loosely consolidated detrital carbonate sediments etc. I do not mean to ask for a detailed review of all study areas, but a sentence or two to get a sense for the range of settings that you are looking at would be great.

L106ff: I do not quite understand the idea behind the role of recharge area. Could you explain that a bit more? Maybe it already helps if you spell out the way you calculate weathering rates (see above).

L174: I think it has to be "faster".

L176: What do you mean by "finding increasing relief"? I do not follow the sentence well.

L194: Minor point – this start to the sentence seems to evoke that it is surprising to find differences in denudation and weathering or at least that there has to be a discussion of possible reasons. Given there is sediment in the rivers, it seems pretty clear to me that explanation (3) definitely has to apply. The question of whether some other biases could apply seems somewhat separate to me.

L224-225: From the figure, it looks like Ireland has the ration of 1.0 – not southern France as the text suggests.

L226: Do you mean to say "*carbonate* weathering rates are largely […]", or do you mean to apply this to all weathering rates?

L231: What do you mean by "carbonates remain subdued" – do you mean something like relief in carbonate landscapes remains subdued?

L254: I think a citation to e.g. Bufe et al. (2021), Erlanger et al. (2021) or Knapp and Tipper (2022) would be more appropriate here. Inferring a limit to carbonate weathering in Bufe et al. (2022) was done in reference to our previous work. Note also that only Erlanger et al. (2021) look at more carbonate rich rocks. The other studies are global Knapp and Tipper (2022) or focused on siliceous rocks (Bufe et al., 2022; Bufe et al., 2021).

L268 – 270: "[…] studies […] show" needs a citation.

L271: Could add Lague (2014). Also, I got confused when I read "channel steepness – erosion rate relationships" because it doesn't say which way around this is. Lague (2014) has values < 1, but I presume you mean $E=f(k_{sn})$ and not $k_{sn}=f(E)$. Could be worth specifying.

I hope that these comments are helpful and remain with best wishes to the author and the editor. Sincerely, Aaron Bufe

**References**

Bufe, A., Cook, K. L., Galy, A., Wittmann, H., and Hovius, N., 2022, The effect of lithology on the relationship between denudation rate and chemical weathering pathways –

evidence from the eastern Tibetan Plateau: Earth Surf. Dynam., v. 10, no. 3, p. 513-530.

Bufe, A., Hovius, N., Emberson, R., Rugenstein, J. K. C., Galy, A., Hassenruck-Gudipati, H. J., and Chang, J.-M., 2021, Co-variation of silicate, carbonate and sulfide weathering drives CO2 release with erosion: Nature Geoscience, v. 14, no. 4, p. 211-216.

Erlanger, E. D., C. Rugenstein, J. K., Bufe, A., Picotti, V., and Willett, S. D., 2021, Controls on Physical and Chemical Denudation in a Mixed Carbonate-Siliciclastic Orogen: Journal of Geophysical Research: Earth Surface, v. 126, no. 8, p. e2021JF006064.

Gabet, E. J., and Mudd, S. M., 2009, A theoretical model coupling chemical weathering rates with denudation rates: Geology, v. 37, no. 2, p. 151-154.

Hilley, G. E., Chamberlain, C. P., Moon, S., Porder, S., and Willett, S. D., 2010, Competition between erosion and reaction kinetics in controlling silicate-weathering rates: Earth and Planetary Science Letters, v. 293, no. 1–2, p. 191-199.

Hilton, R. G., and West, A. J., 2020, Mountains, erosion and the carbon cycle: Nature Reviews Earth & Environment, v. 1, no. 6, p. 284-299.

Knapp, W. J., and Tipper, E. T., 2022, The efficacy of enhancing carbonate weathering for carbon dioxide sequestration: Frontiers in Climate, v. 4.

Lague, D., 2014, The stream power river incision model: evidence, theory and beyond: Earth Surface Processes and Landforms, v. 39, no. 1, p. 38-61.

West, A. J., 2012, Thickness of the chemical weathering zone and implications for erosional and climatic drivers of weathering and for carbon-cycle feedbacks: Geology, v. 40, no. 9, p. 811-814.

West, A. J., Galy, A., and Bickle, M., 2005, Tectonic and climatic controls on silicate weathering: Earth and Planetary Science Letters, v. 235, no. 1–2, p. 211-228.

---

## Author Comment (AC1)

This work presents a nice summary of carbonate weathering and denudation using a compilation of 36Cl rates from around the world. Overall, the methods are sound and the results and discussion are interesting.

We thank the reviewer for this helpful feedback.

One particularly interesting result is that bedrock denudation and weathering rates inferred from rivers roughly agree, even though the timescale integrated by these measurements is quite different. The authors argue that this suggests that weathering dominates denudation of exposed bedrock (rightfully, I think). Does this also mean that weathering rates (at least on average) have remained relatively steady over the past several thousand years?

Yes, we assume that weathering rates have not changed significantly throughout the Holocene. This is highlighted by Figure 5, where we show the precipitation rates from three different paleo-climate models for the mid-Holocene and LGM. In general, the differences in precipitation throughout the integration time of cosmogenic nuclides do not vary substantially for most samples. Hence, we assume that weathering rates have remained relatively constant, at least throughout the Holocene.

Long-term denudation rates from catchments are faster, and the authors present a straightforward analysis that constrains physical erosion rates from these catchments. Weathering still makes up a substantial portion of the mass flux (much more than in silicate-dominated landscapes), and the authors conclude that carbonate landscapes should generally denude more evenly than silicic ones.

The weaknesses here lie mostly in some overlooked literature that should be included, and in some cases discussed. I'd also like to see a bit more information on the field sites, particularly the ones that make up the bulk of the analysis.

The literature that first partitioned denudation into weathering and erosion in silicic landscapes should be cited here. The approach presented here builds on that body of literature by applying these ideas to carbonates, but the approach itself isn't novel. This is also true of some of the weathering bias work (e.g. Riebe and Granger 2012).

We acknowledge that comparing CRN denudation rates to water chemistry measurements has been applied in silicate-rich locations. In the initial submission, we cited Larsen et al. (2014). In the revised version we added two references into the introduction (Blanckenburg et al., 2004; Dixon and Blanckenburg, 2012). We also added another sentence introducing literature and concepts applied to silicic landscapes. We cite from the revised introduction: "Studies applying a combination of CRN denudation rates and elemental analysis, e.g., measuring concentrations of immobile elements in the bedrock and regolith/saprolite have determined denudation portioning in silicic landscapes (Ferrier et al., 2012; Riebe et al., 2001b, 2003, 2004). However, these combined approaches have not been extended to carbonate landscapes, yet."

We are aware that some of the weathering bias approach used and cited in this study is similar to Riebe and Granger (2013). In the initial submission, we exclusively cited Ott et al. (2022) because the method of using stream water chemistry to correct denudation rates has only been published by Ott et al. 2022. Moreover, Riebe and Granger (2013) and earlier studies discuss the effects of quartz enrichment, and the compiled $^{36}$Cl samples rather suffer from calcite depletion. However, we added a reference to Riebe et al. (2001a) to the introduction because this was the first study to correct the effect of quartz enrichment.

More information on field sites would be really useful, especially the Mediterranean ones that form the basis of much of the analysis. Is the bedrock ALL carbonate, or a mix of sedimentary lithologies? How big are the catchments?

This is a good point, and we have included this information at the beginning of the methods section. We cite from the newly added paragraph:

"Published bedrock denudation rates exist for Eastern Asia and the Mediterranean spanning a range of climate zones from the arid Negev desert in Israel to the humid mountains of Japan, with mean annual precipitation ranging between 190 and 2300 mm/a. Alluvial catchments span a narrower range of precipitation rates between 200 and 1100 mm/a and drainage areas ranging between a few km² in southern France to tens of km² in Israel and Crete, Greece (see Fig. S1 for catchment maps). Most of the samples in the compilation are from relatively pure, un- to moderately metamorphosed, massive, and bedded limestone bedrock (Avni et al., 2018; Godard et al., 2016; Ott et al., 2019b; Ryb et al., 2014a, 2014b; Thomas et al., 2017b, 2018; Xu et al., 2013). However, the bedrock samples also contain marbles (Matsushi et al., 2010), pure dolostones and chalk (Ben-Asher et al., 2021), and samples of unknown carbonate lithology (Yang et al., 2020)."

I find the argument that "most weathering happens near the surface" in a highly-reactive lithology, even in the presence of caves, to be unconvincing. Does the correction method of Ott et al 2022 include mass loss by weathering at depths > the attenuation lengthscale? Literature on the silicic rock community has identified deep weathering (which isn't "seen" by cosmogenic nuclides) as potential complication (e.g., Dixon et al., 2009; Campbell et al., 2022). If the correction accounts for deep weathering, it should at least be described in the supplemental material. I actually think it should be addressed briefly in the main text, but I understand that the authors may choose not to devote much space to it if it's covered in the previous publication. The summary in the supplemental material mentions changes in residence time due to differential mineral weathering (e.g. where quartz and carbonates are present together), but doesn't address weathering at depth.

Cosmogenic nuclides cannot account for mass loss below the attenuation length without adding elemental measurements from different depths. Therefore, our approach does not include deep weathering. In contrast to the silicic rock locations mentioned by the reviewer, the carbonate community assumes that due to the fast reaction kinetics, most of the carbonate dissolution occurs close to the Earth's surface. In the initial submission of this manuscript, we provided two references (Gunn, 1981; Worthington and Smart, 2004) with only minor context. To make a

more convincing case and lay out the data that led to this conclusion, we have expanded this part of the method section. We cite from the revised manuscript:

"Erosion can be assumed to equal denudation minus weathering because, despite deep solution features such as caves, volumetrically, most carbonate dissolution occurs close to Earth's surface (Ford and Williams, 2010; Gunn, 2013). Field studies measuring water chemistry at different depths below the surface show that the most of weathering occurs within the first meters below the Earth's surface (Gunn, 1981; Williams and Dowling, 1979). Furthermore, studies quantifying the volumetric percentage of voids in carbonate bedrock found that only 0.003-0.5 % of the karst volume has been removed by deep dissolution (Worthington and Smart, 2004). Hence, we assume weathering rates primarily reflect near-surface mass removal."

Recent work from Erlanger et al (2021) should absolutely be cited here, and their findings should be considered in the discussion. They found a large fraction of dissolved load was actually re-precipitating as carbonate sand. If this were also happening in the catchments studied here, might the measured dissolved loads actually be a minimum estimate of weathering rates from rivers?

This is a good suggestion. The Mediterranean streams from which alluvial samples were taken are all ephemeral. In previous work by Ott et al. (2019), no secondary calcite coating was found in the alluvial samples. Due to similar ephemeral discharge at the sites in southern France and Israel, we assume that the precipitation of secondary calcite does not play a major role there either. However, it is an interesting point that the precipitation of secondary carbonates in wetter climates might effectively limit the amount of weathering export. We added to the discussion of the revised manuscript: "Additionally, Erlanger et al. (2021) showed that significant amounts of dissolved load could reprecipitate as secondary calcite in rivers flowing through carbonates, as soil pore water equilibrated to high $CO_2$ levels degasses upon entering streams. This effect has not been observed in the ephemeral catchments studied here but shows how in wetter regions erosion and weathering can be linked, and catchment and local scale weathering rates may differ."

It seems odd to add an example from outside the study (Ireland) in the final figure. I understand that the authors are trying to provide a low-relief end-member, and I recognize that there simply aren't that many places where 36Cl has been measured in catchments to compare to. Still, I suggest sticking to data reported/analyzed elsewhere in the paper to avoid confusion, rather than bringing in a new setting at the end.

We have removed the data point from Ireland from the revised manuscript.

Small tweaks for clarity:

Fig. 1: It looks like the sites from Ott et al 2019 are just catchments in the main figure, but like they're both catchment and bedrock in the inset.

We thank the reviewer for noticing this mistake and have fixed it in the revised figure.

I'd love to see a map with catchments in the supplemental – it's difficult to assess the size, gradient, topography, etc. when they're only reported as points at the sample location. At the very least, add catchment areas to the table.

We have added a map of the catchments for the three different regions in the supplement.

[Figure]

*Fig. S1: Overview of catchments sampled for catchment-average denudation rates in southern France (A), Israel (B), and Crete (C), with nearby bedrock 36Cl sampling locations.*

Section 4.4: It's hard to assess this info on these 3 sites without the context of relief or slope. It would be easy to add this info here, so readers don't have to go to the supplemental table to find it.

See reply to comment below.

Fig. 4: Color-code marker for sites (warm-to-cool?) in order of relief (or average slope). The red and orange dots are a bit close together in color tone, which makes them harder to distinguish on the figure. You might use a simpler color bar for the erosion rate gradient, perhaps? What's the dashed line?

The dashed line represents the 1:1 line of erosion and weathering. We have added an explanation to the figure caption. We have also included a coloring of data points by local relief.

[Figure]

Recent references mentioned above:

Campbell et al. (2022). Cosmogenic nuclide and solute flux data from central Cuban rivers emphasize the importance of both physical and chemical mass loss from tropical landscapes. Geochronology, 4, 435–453. https://doi.org/10.5194/gchron-4-435-2022

Erlanger, E. D., Rugenstein, J. K. C., Bufe, A., Picotti, V., & Willett, S. D. (2021). Controls on physical and chemical denudation in a mixed carbonate-siliciclastic orogen. *Journal of Geophysical Research: Earth Surface*, *126*, e2021JF006064. https://doi.org/10.1029/2021JF006064

Blanckenburg, F. von, Hewawasam, T., and Kubik, P.W., 2004, Cosmogenic nuclide evidence for low weathering and denudation in the wet, tropical highlands of Sri Lanka: Journal of Geophysical Research: Earth Surface, v. 109, p. 3008, doi:10.1029/2003JF000049.

Dixon, J.L., and Blanckenburg, F., 2012, Soils as pacemakers and limiters of global silicate weathering: Comptes Rendus Geoscience, v. 344, p. 597–609, doi:10.1016/j.crte.2012.10.012 M4 - Citavi.

Ferrier, K.L., Kirchner, J.W., and Finkel, R.C., 2012, Weak influences of climate and mineral supply rates on chemical erosion rates: Measurements along two altitudinal transects in the Idaho Batholith: Journal of Geophysical Research: Earth Surface, v. 117, p. 2026, doi:10.1029/2011JF002231.

Ford, D., and Williams, P.W., 2010, Karst hydrogeology and geomorphology: Chichester, England; A Hoboken, NJ, John Wiley & Sons, 562 p., http://dx.doi.org/10.1002/9781118684986.

Gunn, J., 2013, 6.7 Denudation and Erosion Rates in Karst, in Shroder, J.F. ed., Treatise on geomorphology, London; Waltham, MA, Academic Press, p. 72- 81 TS- CrossRef, doi:10.1016/B978-0-12-374739-6.00115-9 M4 - Citavi.

Gunn, J., 1981, Limestone solution rates and processes in the Waitomo District, New Zealand: Earth Surface Processes and Landforms, v. 6, p. 427–445, doi:10.1002/esp.3290060504.

Larsen, I.J., Almond, P.C., Eger, A., Stone, J.O., Montgomery, D.R., and Malcolm, B., 2014, Rapid soil production and weathering in the Southern Alps, New Zealand: Science (New York, N.Y.), v. 343, p. 637–640, doi:10.1126/science.1244908 PM - 24436184.

Ott, R.F., Gallen, S.F., and Granger, D.E., 2022, Cosmogenic nuclide weathering biases: corrections and potential for denudation and weathering rate measurements: Geochronology, v. 4, p. 455–470, doi:10.5194/GCHRON-4-455-2022.

Riebe, C.S., and Granger, D.E., 2013, Quantifying effects of deep and near-surface chemical erosion on cosmogenic nuclides in soils, saprolite, and sediment: Earth Surface Processes and Landforms, v. 38, p. 523–533, doi:10.1002/esp.3339.

Riebe, C.S., Kirchner, J.W., and Finkel, R.C., 2004, Erosional and climatic effects on long-term chemical weathering rates in granitic landscapes spanning diverse climate regimes: Earth and Planetary Science Letters, v. 224, p. 547–562, doi:10.1016/j.epsl.2004.05.019.

Riebe, C.S., Kirchner, J.W., and Finkel, R.C., 2003, Long-term rates of chemical weathering and physical erosion from cosmogenic nuclides and geochemical mass balance: Geochimica et Cosmochimica Acta, v. 67, p. 4411–4427, doi:10.1016/S0016-7037(03)00382-X.

Riebe, C.S., Kirchner, J.W., and Granger, D.E., 2001a, Quantifying quart enrichment and its consequences for cosmogenic measurements of erosion rates from alluvial sediment and regolith: Geomorphology, v. 40, p. 15–19, doi:10.1016/S0169-555X(01)00031-9.

Riebe, C.S., Kirchner, J.W., Granger, D.E., and Finkel, R.C., 2001b, Strong tectonic and weak climatic control of long-term chemical weathering rates: Geology, v. 29, p. 511–514, doi:10.1130/0091-7613(2001)029<0511:STAWCC>2.0.CO;2.

Williams, P.W., and Dowling, R.K., 1979, Solution of marble in the karst of the Pikikiruna range, Northwest Nelson, New Zealand: Earth Surface Processes, v. 4, p. 15–36, doi:10.1002/ESP.3290040103.

Worthington, S.R.H., and Smart, C.C., 2004, Groundwater in karst: conceptual models, in Gunn, J. ed., Encyclopedia of Caves and Karst Science, Fitzroy Dearborn, p. 399–401.

---

## Author Comment (AC2)

Review of *Erosion and weathering in carbonate regions reveals climatic and tectonic drivers of carbonate landscape evolution* submitted to EGUsphere

I found this paper to be a clearly written and illustrated compilation of chemical weathering and denudation rates in carbonate landscapes. The data compilation and analysis is timely, and the interesting discussion is clearly relevant to understanding landscape evolution under different tectonic and climate boundary conditions. Overall, the manuscript is in a great shape, and I only have a few comments that I suggest to consider before publication.

We thank the reviewer for the helpful and constructive criticism that helped to improve the previous version of this manuscript.

**Calculation of weathering rate**

It would help me if you could state explicitly (i.e. with equations) how you convert concentrations into "weathering rates" in mm/yr and how you interpret these rates. I presume that you need a density to get a quantity in mm/yr? Did you use densities of just Mg and Ca, or did you consider some form of calcite/dolomite?

We added the equation that was used for the calculation to the revised manuscript. We cite from the revised text:
"The weathering rate W in mm/ka is then calculated by assuming that all $[Ca^{2+}]$ and $[Mg^{2+}]$ (mg/l) in the water are derived from carbonate dissolution and using the water flux from our local runoff calculation (mm/ka) for each catchment, with the following equation

$$(1) \quad W = \frac{\left([Ca] + \frac{[Ca]}{M_{Ca}} * M_{CO_3}\right) * (MAP - AET)}{\rho_{Calcite}} + \frac{\left([Mg] + \frac{[Mg]}{M_{Mg}} * M_{CO_3}\right) * (MAP - AET)}{\rho_{Dolomite}}$$

with $\rho_{Calcite}$ and $\rho_{Dolomite}$ of 2.5 and 2.85 g/cm³, respectively."

On a minor note: To me, a denudation rate (in $L\ T^{-1}$) is an average/effective rate of surface lowering. Does the weathering rate here (in $L\ T^{-1}$) have an equivalent interpretation as a surface lowering rate? Typically, dissolution (or weathering) rates on the mineral scale are defined as a change in concentration per time ($Mol\ L^{-3}\ T^{-1}$). On the landscape scale, the term "weathering rate" is inconsistently used in the literature. In general, multiplying concentrations by runoff yields a weathering flux (in the sense of a flow of mass per unit area per unit time). Comparing denudation and weathering fluxes (in $M\ L^{-2}\ T^{-1}$) would be more intuitive for me – but either one works, as long as the calculations are clear.

We chose to express the weathering rate as $[L\ T^{-1}]$ because cosmogenic nuclide denudation rates comprise most of our data compilation, and the majority of the original studies express the cosmogenically-derived denudation rates as $[L\ T^{-1}]$. However, we added a sentence explaining our reasoning to the methods section:

"We chose to express the weathering rate as a surface lowering rate with units of $[L\ T^{-1}]$ because this is the more common way to quantify denudation rates from cosmogenic nuclides."

**Comparison of weathering and denudation rates**

Not all study areas yield measurements of both bedrock and catchment-averaged denudation rates. I don't think it should be a big issue, but in case there were some biases in climate, lithology or tectonic regimes toward the Mediterranean samples with respect to the other samples, it could affect the distributions of the relative rates. Could you state, how much the relative distributions in Fig 2 would change if only those datasets were considered that have all data constrained?

That is why we show panels (B) and (C) in figure 2. These are the only two areas where all three data types are constrained. In both of these areas, the general trends we discuss are the same as in the total compilation. The figure caption of the initial submission mentions this: "The same plots for southern France and the Soreq catchment in Israel; the only two regions where all three data types are available."

**Discussion of weathering limits**

As far as I understand, the authors suggest that silicate weathering is sensitive to physical erosion (L250) in contrast to carbonate weathering that is limited by climate (L209). This seems a bit simplified and perhaps misleading. Increasingly, models and data suggest that silicate weathering fluxes – at least on the scale of mountain catchments – are often not sensitive to denudation and become limited by climate at relatively moderate denudation rates of $10^{-2} – 10^{-1}$ mm/y (Bufe et al., 2022; Bufe et al., 2021; Gabet and Mudd, 2009; Hilley et al., 2010; West, 2012; West et al., 2005). In turn, where carbonates are a minor component of the rocks and waters remain undersaturated with respect to carbonates, they can be sensitive to denudation, even at very high denudation rates (Bufe et al., 2022; Erlanger et al., 2021). Even where waters are saturated with respect to carbonate, they can be indirectly limited by denudation if denudation increases the acid supply (Bufe et al., 2021; Hilton and West, 2020). I do not think that a detailed discussion of weathering limits is necessarily needed for the paper, but I would like to caution against simplifying the discussion to silicate weathering = denudation sensitive and carbonate weathering = climate sensitive.

The reviewer makes a good point. We intended to highlight the higher sensitivity of silicate-landscape weathering to erosion compared to carbonates. In the revised version, we have modified this part of the discussion to highlight that weathering in silicic and carbonate regions can be sensitive to both erosion and climate, while in most settings, climatic control is more important for carbonates. We cite from the revised text:

"Erosion is thought to accelerate weathering by breaking rocks down and increasing surface area (e.g., Rinder and von Hagke, 2021), resulting in a direct feedback between erosion and weathering. In silicate-rich rocks, this has been shown to be true for low and moderate denudation rates, with a transition to a climate limit at high denudation rates (Bufe et al., 2021; West et al., 2005). However, there are observations that suggest that the feedback between erosion and weathering rate is weak or nonexistent in carbonates. Long-term weathering rates of carbonates in experiments were independent of grain size and texture (Levenson et al., 2015). Additionally, water chemistry measurements that suggest carbonate weathering is

generally limited by water and acid availability (Bufe et al., 2021), except for regions where carbonates are a minor component of the bedrock or denudation increases the acid supply (Bufe et al., 2021; Erlanger et al., 2021)."

**Bedrock versus catchment scale**

In conclusion 1, the authors suggest that weathering is dominant on the bedrock scale – because bedrock denudation rates mirror weathering rates. Perhaps I missed some information in the manuscript, but I do not follow. As far as I understand, there is data on (1) denudation on bedrock and catchment scales and (2) weathering on the catchment scale. The contrast between denudation rates on bedrock and catchment scales was assigned to different erosion processes on these spatial scales (e.g. mass movements). Is it therefore possible to compare weathering on the catchment scale to denudation on the bedrock scale? Perhaps, weathering on the bedrock scale is also slower than on the catchment scale – or maybe it is even higher. I am not sure if these rates can be directly compared – or perhaps I missed some discussion of this in the manuscript?

This is an interesting discussion point. Since the weathering rates are calculated on a catchment scale, it is not straightforward to compare them to the bedrock denudation rates from CRNs. In the original submission, we cited several micro erosion meter observations to support the argument that the similarity of bedrock denudation and weathering rates is not a coincidence ("This has also been suggested based on the observations of outcrops with micro erosion meters (Cucchi et al., 1996; Furlani et al., 2009)."). We assume that weathering is distributed relatively uniformly within the landscape. Therefore, it makes conceptual sense to assume that the bedrock rates are similar to weathering rates because weathering is the dominant denudation process at this scale. However, we acknowledge that due to the difference in scale and lack of data on actual weathering distribution, this is a weak argument.
We have added a sentence to acknowledge this weakness in the discussion: "But due to the difference in the spatial scale of catchment weathering rates and local bedrock denudation rates, this link remains vague." We also changed the wording in the conclusion in response to this insightful comment.

**Line comments**

SI: Not all data table entries have units – and not all missing units are obvious. Would be great if fixed.

We have added units to SI table variables with missing units.

L13: "Their" in this sentence refers to "carbonate-dominated landscapes" as far as I understand. What is meant by "their climate sensitivity". Vegetation, denudation, weathering, or other parameters could all be climate sensitive in the landscape.

We have changed this to "climate sensitivity to weathering".

L17: Which discrepancy is "This" referring to? Discrepancy between bedrock denudation versus catchment denudation or discrepancy between catchment denudation and catchment weathering?

We have changed this sentence for clarification.

L19: I do not follow how a lower discrepancy between weathering and denudation makes denudation more spatially uniform? In other words – how does a difference between two rates bear implications for the spatial distribution of these rates?

The inherent assumption is that weathering rates across a landscape are more uniform on a catchment scale than physical erosion.

L32: To me, the word "which" seems to refer to erosion – but it should be denudation?

We have changed this sentence to: "In most silicate-rich landscapes, denudation, the sum of erosion and weathering, is dominated by erosion processes (Larsen et al., 2014)."

L65: At the beginning of the methods, it would be great to have an introductory sentence or two on the selected study areas. What range of denudation rates and climate do they span? Were all available $^{36}$Cl areas included or were some measurements/study areas discarded? Perhaps most importantly, it would be great to have a sense of the range of lithologies. Are these all pure carbonate landscapes, or are there mixed-siliceous rocks. Are the carbonates all unmetamorphosed massive limestones, or are there marbles or more loosely consolidated detrital carbonate sediments etc. I do not mean to ask for a detailed review of all study areas, but a sentence or two to get a sense for the range of settings that you are looking at would be great.

This is a good point, and we have included this information at the beginning of the methods section. We cite from the newly added paragraph:

"Published bedrock denudation rates exist for Eastern Asia and the Mediterranean spanning a range of climate zones from the arid Negev desert in Israel to the humid mountains of Japan, with mean annual precipitation ranging between 190 and 2300 mm/a. Alluvial catchments span a narrower range of precipitation rates between 200 and 1100 mm/a and drainage areas ranging between a few km² in southern France to tens of km² in Israel and Crete, Greece (see Fig. S1 for catchment maps). Most of the samples in the compilation are from relatively pure, un- to moderately metamorphosed, massive, and bedded limestone bedrock (Avni et al., 2018; Godard et al., 2016; Ott et al., 2019b; Ryb et al., 2014a, 2014b; Thomas et al., 2017b, 2018; Xu et al., 2013). However, the bedrock samples also contain marbles (Matsushi et al., 2010), pure dolostones and chalk (Ben-Asher et al., 2021), and samples of unknown carbonate lithology (Yang et al., 2020)."

L106ff: I do not quite understand the idea behind the role of recharge area. Could you explain that a bit more? Maybe it already helps if you spell out the way you calculate weathering rates (see above).

The recharge areas refer to the groundwater catchments of springs. We rephrased the sentence for clarity: "For groundwater water samples, we estimated a recharge area, equivalent to a groundwater catchment, for runoff averaging based on surface topography and local geology (see Fig. S1 for estimated groundwater recharge areas)."

L174: I think it has to be "faster".

We have corrected this mistake.

L176: What do you mean by "finding increasing relief"? I do not follow the sentence well.

The explanation for the increasing relief argument is in the two sentences before. We cite from the original submission:

"Analogous to our findings, a discrepancy between bedrock and catchment-average denudation rates has also been documented in a compilation of silicate-rich rock units (Portenga and Bierman, 2011). This difference has been interpreted as reflecting growing topographic relief because rivers are thought to erode faster than the ridgelines (Hancock and Kirwan, 2007; Small and Anderson, 1998; Thomas et al., 2017)".

L194: Minor point – this start to the sentence seems to evoke that it is surprising to find differences in denudation and weathering or at least that there has to be a discussion of possible reasons. Given there is sediment in the rivers, it seems pretty clear to me that explanation (3) definitely has to apply. The question of whether some other biases could apply seems somewhat separate to me.

The reviewer is right in the sense that a substantial amount of carbonate literature assumes that mechanical erosion at the catchment scale is negligible. We have cited some of these studies in the introduction: "Several recent studies assumed that carbonate erosion is negligible even in mountainous landscapes (Ryb et al., 2014a, 2014b; Avni et al., 2018), implying that weathering dominates carbonate denudation budgets."

Some studies use the dominance of weathering as a definition of karst landscapes. We cite the recent Karst Geomorphology Treatise on Geomorphology issue: "Karst terrains develop where chemical dissolution dominates over mechanical processes." (Frumkin, 2013)

L224-225: From the figure, it looks like Ireland has the ration of 1.0 – not southern France as the text suggests.

The text discusses the ratios of erosion to weathering. The ratio that the reviewer is referring to is between weathering and denudation.

L226: Do you mean to say "*carbonate* weathering rates are largely […]", or do you mean to apply this to all weathering rates?

We have changed this to "carbonate weathering rates".

L231: What do you mean by "carbonates remain subdued" – do you mean something like relief in carbonate landscapes remains subdued?

Yes, we have modified this sentence to "topographically subdued".

L254: I think a citation to e.g. Bufe et al. (2021), Erlanger et al. (2021) or Knapp and Tipper (2022) would be more appropriate here. Inferring a limit to carbonate weathering in Bufe et al. (2022) was done in reference to our previous work. Note also that only Erlanger et al. (2021) look at more carbonate rich rocks. The other studies are global Knapp and Tipper (2022) or focused on siliceous rocks (Bufe et al., 2022; Bufe et al., 2021).

We have changed the citation to (Bufe et al., 2021; Erlanger et al., 2021).

L268 – 270: "[…] studies […] show" needs a citation.

The reviewer refers to the opening statement of the argument, where the following two sentences contain the findings and references (3 references for silicic and 4 for carbonate landscapes). Therefore, we did not add any references to the opening statement.

L271: Could add Lague (2014). Also, I got confused when I read "channel steepness – erosion rate relationships" because it doesn't say which way around this is. Lague (2014) has values < 1, but I presume you mean $E=f(k_{sn})$ and not $k_{sn}=f(E)$. Could be worth specifying.

We have revised the text to refer to the exponent *n,* which clarifies that the relationship referred to is $E = K * k_{sn}^{n}$.

I hope that these comments are helpful and remain with best wishes to the author and the editor. Sincerely, Aaron Bufe

We thank Dr. Aaron Bufe for the thorough review and the important comments and suggestions. We believe that our manuscript has improved thanks to such comments and suggestions.

**References**

Bufe, A., Cook, K. L., Galy, A., Wittmann, H., and Hovius, N., 2022, The effect of lithology on the relationship between denudation rate and chemical weathering pathways – evidence from the eastern Tibetan Plateau: Earth Surf. Dynam., v. 10, no. 3, p. 513-530.

Bufe, A., Hovius, N., Emberson, R., Rugenstein, J. K. C., Galy, A., Hassenruck-Gudipati, H. J., and Chang, J.-M., 2021, Co-variation of silicate, carbonate and sulfide weathering drives CO2 release with erosion: Nature Geoscience, v. 14, no. 4, p. 211-216.

Erlanger, E. D., C. Rugenstein, J. K., Bufe, A., Picotti, V., and Willett, S. D., 2021, Controls on Physical and Chemical Denudation in a Mixed Carbonate-Siliciclastic Orogen: Journal of Geophysical Research: Earth Surface, v. 126, no. 8, p. e2021JF006064.

Gabet, E. J., and Mudd, S. M., 2009, A theoretical model coupling chemical weathering rates with denudation rates: Geology, v. 37, no. 2, p. 151-154.

Hilley, G. E., Chamberlain, C. P., Moon, S., Porder, S., and Willett, S. D., 2010, Competition between erosion and reaction kinetics in controlling silicate-weathering rates: Earth and Planetary Science Letters, v. 293, no. 1–2, p. 191-199.

Hilton, R. G., and West, A. J., 2020, Mountains, erosion and the carbon cycle: Nature Reviews Earth & Environment, v. 1, no. 6, p. 284-299.

Knapp, W. J., and Tipper, E. T., 2022, The efficacy of enhancing carbonate weathering for carbon dioxide sequestration: Frontiers in Climate, v. 4.

Lague, D., 2014, The stream power river incision model: evidence, theory and beyond: Earth Surface Processes and Landforms, v. 39, no. 1, p. 38-61.

West, A. J., 2012, Thickness of the chemical weathering zone and implications for erosional and climatic drivers of weathering and for carbon-cycle feedbacks: Geology, v. 40, no. 9, p. 811-814.

West, A. J., Galy, A., and Bickle, M., 2005, Tectonic and climatic controls on silicate weathering: Earth and Planetary Science Letters, v. 235, no. 1–2, p. 211-228.

Avni, S., Joseph-Hai, N., Haviv, I., Matmon, A., Benedetti, L., and Team, A., 2018, Patterns

and rates of 103−105 yr denudation in carbonate terrains under subhumid to subalpine climatic gradient, Mount Hermon, Israel: GSA Bulletin, v. 131, p. 899–912, doi:10.1130/B31973.1.

Bufe, A., Hovius, N., Emberson, R., Rugenstein, J.K.C., Galy, A., Hassenruck-Gudipati, H.J., and Chang, J.M., 2021, Co-variation of silicate, carbonate and sulfide weathering drives CO2 release with erosion: Nature Geoscience 2021 14:4, v. 14, p. 211–216, doi:10.1038/s41561-021-00714-3.

Erlanger, E.D., Rugenstein, J.K.C., Bufe, A., Picotti, V., and Willett, S.D., 2021, Controls on Physical and Chemical Denudation in a Mixed Carbonate-Siliciclastic Orogen: Journal of Geophysical Research: Earth Surface, v. 126, p. e2021JF006064, doi:10.1029/2021JF006064.

Frumkin, A., 2013, New Developments of Karst Geomorphology Concepts: Treatise on Geomorphology, v. 6, p. 1–13, doi:10.1016/B978-0-12-374739-6.00112-3.

Ryb, U., Matmon, A., Erel, Y., Haviv, I., Benedetti, L., and Hidy, A.J., 2014a, Styles and rates of long-term denudation in carbonate terrains under a Mediterranean to hyper-arid climatic gradient: Earth and Planetary Science Letters, v. 406, p. 142–152, doi:10.1016/j.epsl.2014.09.008.

Ryb, U., Matmon, A., Erel, Y., Haviv, I., Katz, A., Starinsky, A., Angert, A., and Team, A., 2014b, Controls on denudation rates in tectonically stable Mediterranean carbonate terrain: GSA Bulletin, v. 126, p. 553–568, doi:10.1130/B30886.1.